# Intravesical Prostatic Protrusion and Prognosis of Non-Muscle Invasive Bladder Cancer: Analysis of Long-Term Data over 5 Years with Machine-Learning Algorithms

**DOI:** 10.3390/jcm10184263

**Published:** 2021-09-20

**Authors:** Junghoon Lee, Min Soo Choo, Sangjun Yoo, Min Chul Cho, Hwancheol Son, Hyeon Jeong

**Affiliations:** Department of Urology, Seoul Metropolitan Government Boramae Medical Center, Seoul National University College of Medicine, Seoul 07061, Korea; deftblow@gmail.com (J.L.); snuhuro@gmail.com (M.S.C.); ebend@naver.com (S.Y.); cmc1206@empal.com (M.C.C.); volley@snu.ac.kr (H.S.)

**Keywords:** intravesical prostatic protrusion, benign prostatic hyperplasia, bladder cancer, recurrence, machine learning

## Abstract

We aim to investigate the significance of intravesical prostate protrusion (IPP) on the prognosis of non-muscle invasive bladder cancer (NMIBC) after the transurethral resection of bladder tumors (TURBT). For newly diagnosed NMIBC, we retrospectively analyzed the association between prognosis and IPP for at least a 5-year follow-up. A degree of IPP over 5 mm in a preoperative CT scan was classified as severe. The primary endpoint was recurrence-free survival, and the secondary endpoint was progression-free survival. The machine learning (ML) algorithm of a support vector machine was used for predictive model development. Of a total of 122 patients, ultimately, severe IPP was observed in 33 patients (27.0%). IPP correlated positively with age, BPH, recurrence, and prognosis. Severe IPP was significantly higher in the recurrence group and reduced in the recurrence-free survival group (*p* = 0.038, *p* = 0.032). Severe IPP independently increased the risk of intravesical recurrence by 2.6 times. The addition of IPP to the known oncological risk factors in the prediction model using the ML algorithm improved the predictability of cancer recurrence by approximately 6%, to 0.803. IPP was analyzed as a potential independent risk factor for NMIBC recurrence and progression after TURBT. This anatomical feature of the prostate could affect the recurrence of bladder tumors.

## 1. Introduction

Intravesical prostatic protrusion (IPP) is a phenomenon in which the prostate adenoma, mainly the median lobe, grows into the bladder along the least resistant plane, which can occupy the bladder space at a measurable capacity [1]. IPP may be a better noninvasive predictor of bladder outlet obstruction than routine prostatic volume [2]. While lateral bilobular hyperplasia of the prostate can cause compression of the prostatic urethra, IPP of the median lobe may trigger a “ball-valve” type of obstruction, distorting the funneling effect of the normal prostatic-urethral angle and disrupting laminar flow at the bladder neck, leading to dyskinetic movement of the bladder neck during micturition [3]. In addition, a strong bladder contraction force to open a channel between the lobes can aggravate the “ball-valve” effect in IPP [4].

Bladder cancer is currently the second most prevalent form of urinary tract cancer [5]. Up to 80% of patients with non-muscle invasive bladder cancer (NMIBC) experience recurrence, of whom 20% to 30% progress to a higher stage or grade [6]. Two traditional theories have been proposed to explain the frequent occurrence of urothelial tumor multifocality: clonal expansion versus field cancerization [7]. A high degree of IPP has been associated with a high postvoid residual urine volume (PVR) [1]. High PVR is an unfavorable prognostic factor in both theories, due to promoting intraluminal spread and secondary implantation, or due to prolonged exposure to carcinogens as a result of incomplete emptying and subsequent urine stasis [8]. 

Age is a well-known risk factor for bladder cancer [9]; and aging, benign prostatic hyperplasia (BPH), PVR, and IPP have been found to be highly correlated with each other [10,11,12,13]. Thus, aging-induced BPH and lower urinary tract dysfunction may affect the prognosis of bladder cancer. Further investigations are needed to clarify the effects of aging, BPH, PVR, and IPP on the prognosis of bladder cancer.

Over the past few years, the field of artificial intelligence has moved from largely theoretical studies to real-world applications [14]. Recently, there have been attempts to use machine learning (ML), a field of artificial intelligence, in medical research [15]. A support vector machine (SVM) is a supervised ML technique that can be applied to classification or regression analysis [16]. The SVM algorithm tries to construct an optimal separating hyperplane that separates two classes with a maximum margin, where the margin is the largest distance to the nearest training data point of any class. Unlike traditional statistical methods that express SVMs with a single linear formula, SVMs are an attractive approach to data modeling to determine the optimal performance conditions using nonlinear curves placed halfway between the two classes.

An IPP, which can be easily confirmed by a CT scan, is a finding that can suggest BPH and increased PVR. For bladder cancer patients, a CT scan is performed as a perioperative staging work-up test, but PVR tests are not routinely performed. Therefore, it would be meaningful to analyze the association between bladder cancer and IPP on preoperative CT scans. In this study, we investigated the effect of IPP on the prognosis of NMIBC after transurethral resection of bladder tumors (TURBT) through analysis of long-term data, over 5 years, with ML algorithms. We also evaluated the correlation between IPP, aging, and BPH.

## 2. Materials and Methods

### 2.1. Ethics Approval and Consent

This study was approved by our local institutional review board (IRB), the Seoul National University-Seoul Metropolitan Government Boramae Medical Center (IRB No. 10-2021-15). Because this study retrospectively analyzed anonymized data, informed consent was waived by the Seoul National University-Seoul Metropolitan Government Boramae Medical Center. All study protocols were conducted in compliance with the principles of the Declaration of Helsinki guidelines

### 2.2. Study Design

After approval by our local institutional review board (10-2021-15), electronic charts of consecutive male patients who underwent TURBT for bladder tumors were reviewed between August 2012 and July 2015. Based on the inclusion and exclusion criteria, patients with pathologically confirmed, newly diagnosed primary NMIBC with a minimum follow-up of 5 years and eligible preoperative CT scans were included, and those with invasive bladder tumors (stage T2 or greater) at diagnosis, upper tract primaries, or previous BPH surgery were excluded.

Clinicopathological variables included patient demographics and all parameters used in the previous EORCT or CUETO risk models, such as smoking history, cytology results, associated CIS, pathological stage and grade, tumor size, and multiplicity, as well as prostate volume and presence of IPP, with its grade [17,18,19]. Tumor size was defined as the largest diameter of the surgical specimen on macroscopic analysis. We used the size of the largest tumor as the standard in cases of multiple bladder tumors. Clinicopathological parameters were evaluated according to the 2016 WHO classification system [20]. We defined presence of CIS, T1 high-grade (HG), Ta low-grade (LG) with 3 risk factors, Ta HG with at least 2 risk factors, or T1 LG with at least 2 risk factors as high-risk NMIBC. Low-risk NMIBC was defined as Ta LG with no or at most one risk factor. All others were classified as intermediate-risk NMIBC. The additional clinical risk factors considered were: age > 70; multiple papillary tumors; and tumor diameter > 3 cm [21].

IPP is the distance in millimeters, measured from a magnified image of the prostate obtained from preoperative CT urography in the midline sagittal view by drawing a perpendicular line from the anterior to posterior intersections of the bladder base and the innermost tip of the protrusion [22]. The degree of protrusion was classified as grade I (5 mm or less), grade II (greater than 5 to 10 mm), or grade III (greater than 10 mm). We defined IPP of grades 2 and 3 as severe IPP. 

We performed follow-up cystoscopy and urine cytology every 3 months during the first 2 years after surgery, every 6 months during the next 3 years, and once yearly thereafter. Appropriate upper tract evaluation via CT scan was obtained annually. A second-look TURBT was recommended routinely in all cases of pathologically HG T1 tumors. Even if cancer appeared as a result of the second-look TURBT, it was not judged as a recurrence. BCG therapy was recommended if tumors were pathologically diagnosed as HG or concomitant CIS.

The primary endpoint was recurrence-free survival (RFS), defined as the time from first treatment to the first recurrence. The secondary endpoint was progression-free survival (PFS), defined as the time from the first treatment to the first progression. To analyze the effect of IPP on the prognosis of NMIBC, we analyzed RFS and PFS for all patients, and then further identified the effect of IPP on high-risk NMIBC patients. Tumor recurrence was indicated for cases identified as Ta, T1, or CIS in the histopathological examination after TURBT for suspected recurrent lesions. Tumor progression was defined as the development of muscle invasion or distant metastasis. Patients who were still alive and without recurrence or progression were censored at the date of the last available follow-up cystoscopy.

### 2.3. Statistical Analysis

Descriptive statistical analysis was conducted using an independent-sample *t*-test for continuous variables or a chi-square test for categorical variables between the two groups.

Survival curve analyses for time to tumor recurrence and progression were performed using the Kaplan–Meier method, and the differences between curves were analyzed by the log-rank test. Multivariate Cox proportional hazard regression models using the stepwise backward selection method were performed to identify significant independent predictors for 60-month recurrence and progression among covariates. Multivariate analysis was used to estimate the hazard ratio for each clinicopathological factor.

The ML algorithm of the SVM was used for further analysis and predictive model development. First, the correlation between each factor was identified with a correlation heat map, and the correlation coefficient of each factor was evaluated using the logistic regression algorithm. Next, using the SVM algorithm, we calculated the contribution of each factor in predicting prognosis. The prediction accuracy of each algorithm was expressed as an F1-score, which is the harmonic mean of the precision and recall. The area under the receiver operating characteristic curve (AUC) value was also calculated.

Statistical analysis was performed using SPSS^®^ (version 26.0, SPSS, IBM company, Armonk, NY, USA), Python (Python Language Reference version 3.7, Python Software Foundation, Beaverton, OR, USA), and REX (version 1.0, Rexsoft, Seoul, Korea) statistical software. All statistical tests were 2-sided and considered to be significant if the *p*-value was <0.05. Hazard ratios were estimated with 95% confidence intervals (CIs).

## 3. Results

### 3.1. Study Patients 

A total of 518 patients received TURBT for bladder tumors for 3 years in our institution. The total number of patients in the initial case series was 327, excluding duplicate surgical cases. Of these, 81 female patients (15.5%) were excluded, followed by 122 patients who met the exclusion criteria (26 due to benign bladder disease, 8 due to early cystectomies, 21 due to upper tract primary disease, 8 due to previous or concomitant transurethral resection of the prostate, 18 due to stage T2 or greater disease, 23 due to loss to follow-up, and 18 due to incomplete records). A total of 122 patients who received initial TURBT with NMIBC and were followed for more than 5 years were ultimately analyzed.

### 3.2. Differences in IPP According to Bladder Cancer Recurrence

Patient and tumor characteristics are presented in Table 1. The median (quartile) age was 71.0 (64.3–76.8) years. The median prostate volume of the included patients was 28.6 (20.8–35.9) mL, and 58 patients (47.5%) had clinically significant BPH over 30 mL. IPP was observed in 54 patients (44.3%), and severe IPP above grade 2 was observed in 33 patients (27.0%) (grade 2: 26 (21.3%); grade 3: 7 (5.7%)). There was no difference in the presence of BPH between the recurrence and non-recurrence groups (recurrence group vs. non-recurrence group = 51.4% vs. 41.7%, *p* = 0.238). However, the rate of severe IPP of grade 2 or higher was significantly higher in the recurrence group than in the non-recurrence group (recurrence group vs. non-recurrence group = 33.8% vs. 16.7%, *p* = 0.038). Correlation heatmaps show that IPP correlated positively with age, BPH, recurrence, and prognosis (Figure 1a). Age and prostate volume gradually increased with IPP grade (Figure 1b,c). In addition, IPP was analyzed as a risk factor for intravesical recurrence, even after adjusting for other known risk factors, such as age, smoking history, urine cytology, and stage (Figure 1d).

### 3.3. Effect of IPP on Prognosis of Bladder Cancer

The median (quartile) follow-up was 36 (7–70) months, namely, 9 (5–23) months in the recurrence group and 72 (69–79.5) months in the non-recurrence group. Disease recurrence occurred in 74 patients (60.7%) and progression occurred in 11 patients (9.0%) during the follow-up period. The 1-, 3-, and 5-year RFS rates were 63.1%, 49.2%, and 40.2%, respectively, and the 1-, 3-, and 5-year PFS rates were 96.8%, 91.6%, and 86.1%, respectively. RFS rates were significantly different with the presence or absence of severe IPP grade (grades 2–3) in the Kaplan–Meier analysis (log-rank test, *p* = 0.032) (Figure 2a). Severe IPP did not show statistical significance for increased risk of progression (log-rank *p* = 0.069) (Figure 2b). In patients with high-risk NMIBC, severe IPP has been shown to significantly reduce both RFS and PFS (*p* = 0.034 and *p* = 0.009, respectively) (Figure 2c,d). However, severe IPP did not significantly affect the recurrence and progression in low- to moderate-risk NMIBC patients. Grades 2–3 IPP of 5 mm or more independently increased the risk of intravesical recurrence by 2.6 times after initial TURBT (Table 2).

In the prediction model using the SVM algorithm, the stage achieved the highest accuracy of 0.6523 among single factors, and the accuracy gradually increased as each factor was added. Using all factors, the best accuracy of 0.803 and an AUC of 0.749 were shown. When predicting recurrence with already known risk factors, the best-balanced accuracy was 0.754, but when IPP was added to the existing factors, the prediction accuracy increased by 6%, to 0.803 (Figure 3)

## 4. Discussion

The results of this study confirmed that IPP affects the prognosis of NMIBC. In particular, severe IPP had an adverse effect on both RFS and PFS in high-risk NMIBC patients. Severe-grade IPP (grade 2 or 3 lesions, 5 mm or larger) increased the risk of intravesical recurrence after TURBT by 2.6 times and had a tendency to progress to T2. The addition of IPP to the known histopathological and oncological risk factors in the prediction model improved the predictability of bladder cancer recurrence by approximately 6%. To the best of our knowledge, this study is the first to confirm that IPP could influence the prognosis of bladder cancer. In male patients with bladder cancer, lower urinary tract dysfunction should also be evaluated, and active treatment should be considered.

IPP independently increased the risk of intravesical tumor recurrence by 2.6 times. Urodynamically, it is known that IPP increases PVR through the ball-valve effect and distortion of the prostato-urethral angle [1]. According to the hypothesis, incomplete emptying and urine stasis can accelerate bladder carcinogenesis by promoting intraluminal implantation of floating cancer cells and/or prolonging the exposure to carcinogens [8]. An experimental animal study found that the incidence of superficial bladder carcinoma was greater in a partial bladder outlet obstruction-induced rat model [23]. Early obstruction removal of simultaneous TURBT with transurethral resection of the prostate was associated with decreased rates of recurrence in the long term in one human observational study [6]. In addition, prolonged exposure to a carcinogen and implantation of floating cancer cells due to urine retention could activate the inflammatory response of patients. Recent studies showing that elevated C-reactive protein or pyuria, a result of an increased inflammatory response, are useful in predicting poor prognosis of bladder cancer and may also be related to these hypotheses [24,25]. 

Rigid cystoscopy has a structural limitation whereby access to the bladder neck and trigone area are not easy. IPP severely worsens the visual field and movement of rigid cystoscopy [1]. Fukuhara et al. reported that rigid fluorescence cystoscopy was more difficult for diagnosing bladder cancer at the base than flexible fluorescence cystoscopy [26]. The sensitivity, specificity, and false-positive rates of bladder cancer diagnosis of flexible cystoscopy and rigid cystoscopy were 100%, 82.6%, and 17.4%; and 93.4%, 58.8%, and 41.2%, respectively. Therefore, additional difficulties in diagnosis and treatment caused by IPP such as missing mass or incomplete resection may adversely affect the oncological outcome of recurrence and progression. In this study, through long-term follow-up data for a minimum of five years, IPP could affect the prognosis of NMIBC both in the long term and in the short term.

It has been reported that bladder cancer can be considered for aggressive treatment despite being diagnosed in the elderly with a median age of 73 years [27]. The relative risk of recurrence was 1.85 in elderly patients over 70 years old in this study. Age is a well-known risk factor for bladder cancer [28]. As with incidence, recurrence of bladder cancer is also higher in elderly patients [9]. The possible theoretical backgrounds include genetic instability, decreased host immunology, and delayed diagnosis in elderly patients [29]. However, the pathophysiology of the effect of aging on the recurrence of bladder cancer is poorly understood and is explained only by epidemiological evidence. In the present study, it was confirmed that aging, BPH over 40 mL, and IPP are positively correlated. This finding corresponds with earlier studies that reported that aging, BPH, and IPP are highly correlated with each other [10,30]. For a comprehensive prognosis model, further investigations are needed on the effects of aging, lower urinary tract symptoms, prostate volume, and IPP on the prognosis of bladder cancer.

IPP may have a greater effect on LUTS or bladder outlet obstruction than prostate volume. The intravesical hydraulic energy during voiding exerts sufficient pressure to deform the prostate [31]. The anterior prostate is attached to the pubo-prostatic ligaments, the lateral side to the endopelvic fascia, and the posterior side to the Denonvilliers’ fascia [32]. Therefore, IPP protruding towards the bladder is most susceptible to the radial component of intravesical pressure during voiding [1]. Unlike bilateral lobal BPH, in IPP patients, the more force that is applied for urination, the greater the ball-valve effect tends to be [31]. These obstructive effects are mainly associated with a high PVR [1]. Older age and a longer IPP were independent factors associated with the pathogenesis of bladder stone formation and disruption of the excretion of the stone nidus [30]. Similarly, urinary stasis is believed to lead to carcinogen exposure and could increase the risk of bladder cancer recurrence [23].

This study also has several limitations. First, we recognize that it was a retrospective study with a small number of patients. However, there were no previous studies on the association between IPP and bladder cancer. We expect researchers to be interested in the reporting of our results on the association between IPP and bladder cancer. Second, there were no data on PVR values in this study. Although a PVR test is simple and easy, it is not routinely performed on bladder cancer patients. We analyzed the data with an interest in IPP found in routinely performed preoperative CT scans. Third, we cautiously suggested that LUTS/BPH and IPP might be one of the causes of age influencing the prognosis of bladder cancer. However, since there are so many confounders, it could be just the result of a general tendency of aging. As this is a subject that has not been studied yet, clearer results can be shown if these limitations are corrected in other studies. 

## 5. Conclusions

Of the patients who underwent TURBT for noninvasive bladder cancer, IPP was analyzed as a potential independent risk factor for bladder cancer recurrence. In addition to the oncological features of bladder cancer itself, it is also important to evaluate the state of LUTS and protrusion of the prostate. These anatomical features of the prostate, particularly the grade of IPP, could affect the recurrence of bladder tumors.

## Figures and Tables

**Figure 1 jcm-10-04263-f001:**
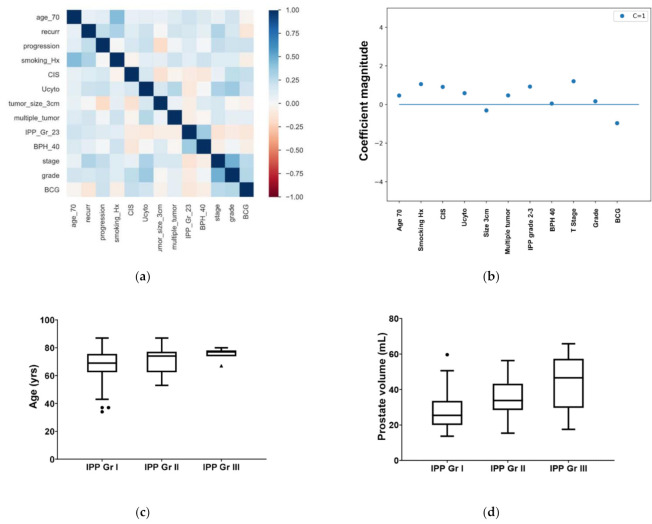
(**a**) Pearson correlation heatmaps between intravesical prostatic protrusion, known 10 risk factors, and recurrence/progression of non-muscle invasive bladder cancer (NMIBC). (**b**) Comparison of coefficient magnitude of each risk factor for recurrence of NMIBC after TURBT in logistic regression analysis. (**c**) Changes in age according to IPP grade (mean ages are shown; error bars represent standard error of the mean). (**d**) Changes in prostate volume according to IPP grade (mean ages are shown; error bars represent standard error of the mean). CIS = carcinoma in situ; BCG = Bacillus Calmette–Guérin; BPH = benign prostatic hyperplasia; IPP = intravesical prostate protrusion.

**Figure 2 jcm-10-04263-f002:**
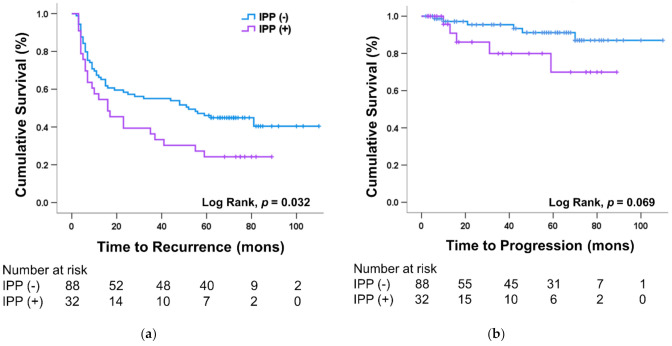
Kaplan−Meier plot for (**a**) recurrence and (**b**) progression comparing patients with or without severe-grade (grade 2–3) intravesical prostatic protruding (IPP). The difference in (**c**) recurrence and (**d**) progression with or without severe IPP in patients with high-risk non-muscle invasive bladder cancer.

**Figure 3 jcm-10-04263-f003:**
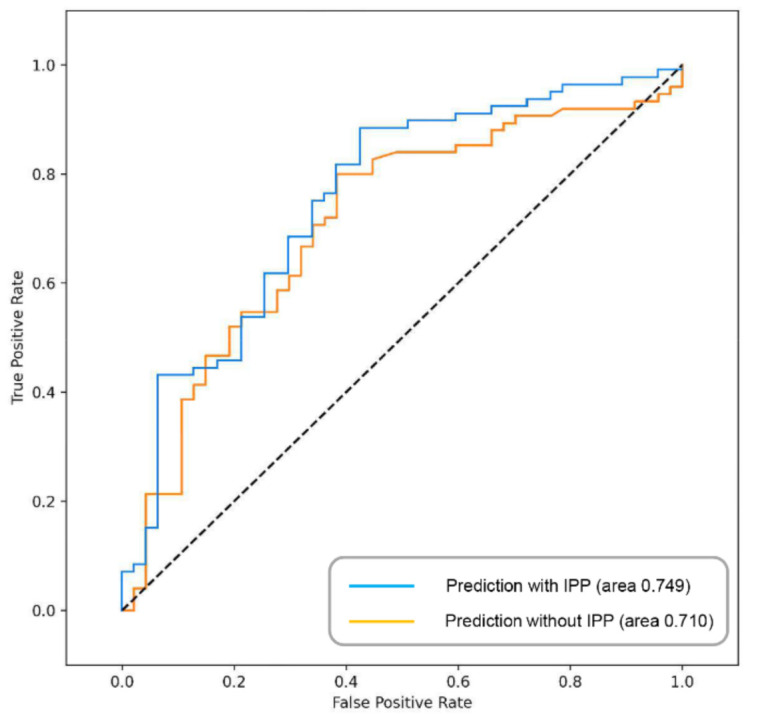
ROC analysis of machine learning prediction model with or without intravesical prostatic protrusion. ROC: receiver operating characteristic.

**Table 1 jcm-10-04263-t001:** Clinical and pathologic characteristics of patients with or without intravesical recurrence.

	Total (*N* = 122)	Recurrence (*n* = 74)	No Recurrence (*n* = 48)	*p*
Age, mean ± SD	69.05 ± 10.08	70.46 ± 8.49	66.88 ± 11.90	0.055
Smoking history	43 (35.2%)	32 (43.2%)	11 (22.9%)	0.017
CIS	7 (5.7%)	6 (8.1%)	1 (2.1%)	0.162
Urine cytology	47 (38.5%)	35 (47.3%)	12 (25.0%)	0.013
Tumor size > 3 cm	31 (25.4%)	19 (25.7%)	12 (25.0%)	0.933
Multiple tumor	61 (50.0%)	42 (56.8%)	19 (39.6%)	0.064
Stage (I/a)	54 (44.3%)/68 (55.7%)	41 (55.4%)/33 (44.6%)	13 (27.1%)/35 (72.9%)	0.002
Grade (HG/LG)	60 (49.2%)/62 (50.8%)	41 (55.4%)/33 (44.6%)	19 (39.6%)/29 (60.4%)	0.088
BCG treatment	48 (39.3%)	26 (35.1%)	22 (45.8%)	0.182
BPH	58 (47.5%)	38 (51.4%)	20 (41.7%)	0.238
IPP (Gr2-3)	33 (27%)	25 (33.8%)	8 (16.7%)	0.038

SD = standard deviation; CIS = carcinoma in situ; HG = high grade; LG = low grade; BCG = Bacillus Calmette–Guérin; BPH = benign prostatic hyperplasia; IPP = intravesical prostate protrusion.

**Table 2 jcm-10-04263-t002:** Stepwise multiple logistic regression analysis for predictive factors of intravesical recurrence in non-muscle invasive bladder cancer after TURBT.

		Univariable			Multivariable	
	OR	95% CI	*p*	OR	95% CI	*p*
Age > 70	1.128	0.710–1.790	0.610	1.930	1.560–5.196	0.018
Smoking history	2.250	1.321–3.834	0.003	2.847	1.560–5.196	0.001
Urine cytology	1.930	1.220–3.054	0.005	1.623	0.996–2.644	0.052
Prostate volume over 30 mL	1.318	0.831–2.088	0.240	1.602	0.919–2.793	0.097
IPP Gr2-3	1.686	1.040–2.734	0.034	2.646	1.425–4.911	0.002
T stage	2.159	1.361–3.424	0.001	2.443	1.451–4.114	0.001
Grade	1.667	1.053–2.640	0.029			
Size > 3 cm	1.058	0.628–1.783	0.833			
Multiplicity	1.523	0.961–2.413	0.073			
CIS	1.813	0.785–4.188	0.164			
BCG	0.775	0.481–1.250	0.296	0.576	0.348–0.952	0.031

OR = odds ratio; CI = confidence interval; IPP = intravesical prostate protrusion; CIS = carcinoma in situ; BCG = Bacillus Calmette–Guérin; TURBT = Transurethral resection of bladder tumor.

## Data Availability

The datasets generated during and/or analyzed during the current study are available from the corresponding author on reasonable request.

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
