# Peer review of "Intravesical Prostatic Protrusion and Prognosis of Non-Muscle Invasive Bladder Cancer: Analysis of Long-Term Data over 5 Years with Machine-Learning Algorithms"

_jcm, 2021, doi:10.3390/jcm10184263_

Round 1
Reviewer 1 Report
IPP also tended to increase the risk of progression - 'This sentence should there was trend towards increased risk of progression.' as it was statistically insignificant
Bottom legend of figure 2b should state month to progression (it say recurrence)
IPP grade 2-3 have 2.6 times increased odds of TURBT recurrence . Current statement reports 2.4 which is not the case as per TABLE 2
Discussion: 'The results could propose another interesting hypothesis that lower urinary tract symptoms (LUTS)/BPH associated with aging could affect bladder cancer prognosis' - I am not sure how are they generating this hypothesis. It is not clear for the reader. BPH did not have statistical significance for recurrence but prostate volume did. At times with smaller number we see spurious correlations. This should be avoided.
Author Response
1. IPP also tended to increase the risk of progression - 'This sentence should there was trend towards increased risk of progression.' as it was statistically insignificant
- We appreciate the reviewer’s good comment. We replaced this sentence to “Severe IPP did not show statistical significance for increased risk of progression (log-rank p=0.069) (Figure 2b).” in line 191 on page 6.
2. Bottom legend of figure 2b should state month to progression (it say recurrence)
- Thanks for your comment. We changed "Time to recurrence" to "Time to Progression”.
3. IPP grade 2-3 have 2.6 times increased odds of TURBT recurrence . Current statement reports 2.4 which is not the case as per TABLE 2
- Thanks for your point We replaced "2.4" to "2.6".
4. Discussion: 'The results could propose another interesting hypothesis that lower urinary tract symptoms (LUTS)/BPH associated with aging could affect bladder cancer prognosis' - I am not sure how are they generating this hypothesis. It is not clear for the reader. BPH did not have statistical significance for recurrence but prostate volume did. At times with smaller number we see spurious correlations. This should be avoided.
- Thank you for your good comment. We removed this sentence from the discussion section to avoid confusion from readers.
We thank you for the invaluable comments and helpful suggestions that contributed to revision of our manuscript.

Reviewer 2 Report
Authors aimed to analyse how intravesical prostatic protrusion could aid in prognosis of non-muscle invasive bladder cancer in a long-term data gathering over 5 years using machine-learning algorithms. The aim of the study is original and I enjoyed reading it. Study design and stats are sound. The aim is clear and executed in accordance with data reporting guidelines. However, I am unsure if IPP alone can be interpreted as prognosticator. Recently, I found an interesting long-term analysis discussing the TNR-C score for patients with bladder cancer (PMID: 32114587). Maybe the authors of the present study also find merit in reading and shortly discussing it. When battling bladder cancer new information on prognosis could be used in the future to construct a comprehensive prognosis model. Also, authors discussed the relation of BPH and age. This is - of course - a given as BPH progresses with age and median age of diagnosis for bladder cancer is 73 years. I hope authors find merit in quickly discussing how to treat the oldest-old patient cohort in light of progressive disease (>/=T2) (PMID: 32871580).
Author Response
1. Recently, I found an interesting long-term analysis discussing the TNR-C score for patients with bladder cancer (PMID: 32114587). Maybe the authors of the present study also find merit in reading and shortly discussing it.
- We appreciate the reviewer’s good comment. Risk factors for the prognosis of bladder cancer are diverse, and it is of course possible that IPP alone does not act directly. We additionally discussed C-reactive protein as an example as a prognostic factor for bladder cancer that reflects the inflammatory response in line 237 on page 9. The following are the description:
“In addition, prolonged exposure to carcinogen and implantation of floating cancer cells due to urine retention could activate the inflammatory response of patients. Recent studies showing that elevated C-reactive protein or pyuria, a result of an increased inflammatory response, are useful in predicting poor prognosis of bladder cancer may also be related to these hypotheses [1, 2].”
1. Tamalunas, A., et al., Impact of Routine Laboratory Parameters in Patients Undergoing Radical Cystectomy for Urothelial Carcinoma of the Bladder: A Long-Term Follow-Up. Urol Int, 2020. 104(7-8): p. 551-558.
2. Suh, J., et al., Pyuria as a Predictive Marker of Bacillus Calmette–Guérin Unresponsiveness in Non-Muscle Invasive Bladder Cancer. Journal of Clinical Medicine, 2021. 10(17): p. 3764.
2. Also, authors discussed the relation of BPH and age. This is - of course - a given as BPH progresses with age and median age of diagnosis for bladder cancer is 73 years. I hope authors find merit in quickly discussing how to treat the oldest-old patient cohort in light of progressive disease (>/=T2) (PMID: 32871580).
- Thank you for your suggestion. BPH, IPP, and age are probably related factors. As you mentioned, age is known to be an important prognosis for bladder cancer, and aggressive treatment can be considered even in advanced age. We descripted as follow in line 253 and 266 on page 10:
“It has been reported that bladder cancer can be considered for aggressive treatment despite being diagnosed in the elderly with a median age of 73 years [3].”
“Further investigations for comprehensive prognosis model are needed on the effects of aging, lower urinary tract symptoms, and IPP to the prognosis of bladder cancer.”
3. Tamalunas, A., et al., Is It Safe to Offer Radical Cystectomy to Patients above 85 Years of Age? A Long-Term Follow-Up in a Single-Center Institution. Urol Int, 2020. 104(11-12): p. 975-981.
We thank you for the invaluable comments and helpful suggestions that contributed to revision of our manuscript.
Reviewer 3 Report
The findings of this paper is interesting and may provide an important contribution to improved accuracy of risk classification of non-muscle-invasive bladder cancer. It would be better if the presence or absence of IPP was incorporated in to the EAU risk group and improved resolution was observed.
Minor
P5 line 184: Grades 2-3 IPP of 5 mm or more independently increased the risk of intravesical recurrence by 2.6, not 2.4.
P6: Figure 2: The title of horizontal axis in Fig. 2(b) could be ”Time to Progression”, not "Time to recurrence"
Author Response
1. It would be better if the presence or absence of IPP was incorporated in to the EAU risk group and improved resolution was observed.
- We appreciate the reviewer’s good comment. We further analyzed the effects of severe IPP on RFS and PFS by EAU risk group using Kaplan-Meier curve analysis. In low- to moderate-risk NMIBC, IPP had no significant effect, but in high-risk NMIBC, the presence of severe IPP has been shown to significantly worsen RFS and PFS (p=0.034 and p=0.009, respectively). We descripted as follow in line 97 on page 3, line 193 on page 6, line 222 on page 9 and figure 2:
“We defined T1 high-grade (HG), presence of CIS, Ta low-grade (LG) with 3 risk factors, Ta HG with at least 2 risk factors, or T1 LG with at least 2 risk factors as high-risk NMIBC. Low-risk NMIBC was defined as Ta LG with no or at most one risk factor. All others were classified as intermediate-risk NMIBC. Additional clinical risk factors of above are: age > 70; multiple papillary tumors; tumor diameter > 3 cm [1].”
“In patients with high-risk NMIBC, severe IPP has been shown to significantly reduce both RFS and PFS (p=0.034 and p=0.009, respectively) (Figure 2c, Figure 2d). However, severe IPP did not significantly affect the recurrence and progression in low- to moderate-risk NMIBC patients.”
“In particular, severe IPP had an adverse effect on both RFS and PFS in high-risk NMIBC patients.”
“Figure 2. Kaplan-Meier plot for (a) recurrence and (b) progression comparing patients with or without severe-grade (grade 2-3) intravesical prostatic protruding (IPP). And the difference in (c) recurrence and (d) progression with or without severe IPP in patients with high-risk non-muscle invasive bladder cancer.”
1. Babjuk, M., et al., EAU Guidelines on Non-muscle-invasive Bladder Cancer. Presented at the EAU Annual Congress Milan 2021. ISBN 978-94-92671-13-4. http://uroweb.org/guidelines/compilations-of-all-guidelines/. 2021.
2. P5 line 184: Grades 2-3 IPP of 5 mm or more independently increased the risk of intravesical recurrence by 2.6, not 2.4.
- Thanks for your point We replaced "2.4" to "2.6".
3. P6: Figure 2: The title of horizontal axis in Fig. 2(b) could be ”Time to Progression”, not "Time to recurrence"
- Thanks for your comment. We changed "Time to recurrence" to "Time to Progression”.
We thank you for the invaluable comments and helpful suggestions that contributed to revision of our manuscript.